# Minimizing Chebyshev Risk Magically Mitigates the Perils of Overfitting

## Abstract

Since reducing overfitting in deep neural networks (DNNs) increases their test performance, many efforts have tried to mitigate it by adding regularization loss terms in one or more hidden layers of the network, including the convolutional layers. To build upon the canonical wisdom guiding these previous works, we analytically tried to understand how intra and inter-class feature relationships affect misclassification. Our analysis begins by assuming a DNN is the composition of a feature extractor and classifier, where the classifier is the last fully connected layer of the network and the feature layer is the input vector to the classifier. We assume that, corresponding to each class, there exists an ideal feature vector which we designate as a class prototype. The goal of the training method is then to reduce the probability that an example's features deviate significantly from its class prototype, which increases the risk of misclassification. Formally, this probability can be bound using a Chebyshev's inequality comprised of within-class covariance and between-class prototype distance. The terms in the inequality are added to our loss function for optimizing the feature layer, which implicitly optimizes the previous convolutional layers' parameter values. We observe from empirical results on multiple datasets and network architectures that our training algorithm reduces overfitting and improves upon previous approaches in an efficient manner.

## 1 Introduction

Deep neural networks (DNNs) have shown to be effective pattern extractors and classifiers, resulting in remarkable and increasing performance in visual classification tasks over the last two decades. A large portion of the performance increase may be attributed to improved architectures, increased scales, and larger quantity training sets, but these classifiers are still at risk to the phenomenon of overfitting to a particular training set, which equates to rote memorization of specific examples and decreased generalization.

Many strategies have been developed over the years to regularize networks during the training process such as data augmentation, weight decay, dropout, and batch normalization, which have now become standard practices in image classifier training. In parallel to these training strategies, several efforts have looked at augmenting the standard cross-entropy loss function with additional terms that seek to decorrelate learned feature representations and eliminate redundant weights.

In our paper, we analyze the mathematical basis for removing the covariance between feature representations and in doing so, transfer the concept of the *class prototype* from the field of deep metric learning (Mensink et al., 2012) into our derivations. We utilize the class prototype, which is a comprehensive set of learned features that represent a class' examples, to derive Chebyshev probability bounds on the deviation of an example from it's class prototype and inspire a new loss function that we empirically show to perform comparatively well to previous efforts. In this paper, we make the following contributions:

- A theoretical framework based on Chebyshev probability bounds under which regularization and related training techniques can be analyzed. The bound admits a

new optimizable metric called Chebyshev Prototype Risk (CPR), which bounds the deviation in similarity between the features of an example and its true prototype.

- We design a new loss function based on our probability bounds that reduces intra-class feature covariance while keeping inter-class feature vectors separated, which reduces the risk of overfitting.

- We provide an algorithm for reducing feature covariance that scales with the number of categories rather than the number of training examples, thus allowing our method to be used on very large datasets effectively.

- We show evidence that minimizing the CPR is a necessary condition to avoid over-fitting.

## 1.1 PRELIMINARY NOTATIONS

There exists a generic classifier $f(\boldsymbol{x}; \theta)$ that maps an input vector $\boldsymbol{x} \in \mathbb{R}^M$ to a vector $\boldsymbol{y} \in \mathbb{R}^K$. We select the parameters, $\theta$, of this classifier as an estimator using a learning algorithm $\mathcal{A}(\mathcal{D})$ that takes as input a labeled training set $\mathcal{D} = \{(\boldsymbol{x}_1, \boldsymbol{y}_1), (\boldsymbol{x}_2, \boldsymbol{y}_2)...(\boldsymbol{x}_N, \boldsymbol{y}_N)\}$. This classifier can be broken down into the composition of a *feature extraction* function $g : \mathbb{R}^M \to \mathbb{R}^J$ and a *feature classification* function $h : \mathbb{R}^J \to \mathbb{R}^K$. Therefore, $\boldsymbol{y} = f(\boldsymbol{x}; \theta) = h(g(\boldsymbol{x}; \theta_g); \theta_h)$, where $\theta_g$ and $\theta_h$ are the disjoint parameter sets on which $g$ and $h$ are dependent, respectively. Unless otherwise stated, the norm $|| \cdot ||$ will refer to the $L_2$-norm.

There exists for each category a learnable, class representative vector $\boldsymbol{p}_k \in \mathbb{R}^J$ that is learned during training and minimizes some feature similarity based loss function between itself and all of its constituent examples in class $k$. Thus, there is a defined set of class prototype feature vectors $\{\boldsymbol{p}_1, ..., \boldsymbol{p}_K\}$.

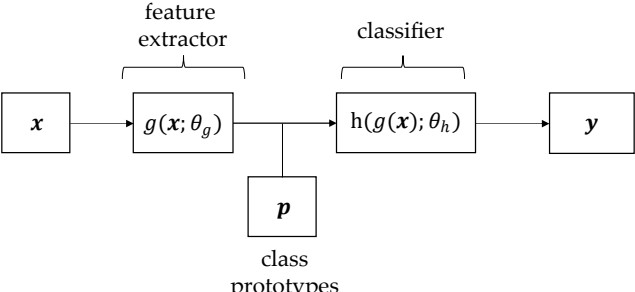

Figure 1: Neural network split into a feature extractor and classifier (last fully connected layer) acting on an input $\boldsymbol{x}$. Class prototypes are learned feature vectors that comprehensively represent each class' trained features.

## 2 RELATED WORK

**DeCov**   The authors of (Cogswell et al., 2016) discourage feature correlations at selected layers of the deep neural network by implementing an explicit penalty loss term on intra-batch covariance. For a given mini-batch, they define the batch feature covariance as,

$$C_{i,j} = \frac{1}{N} \sum_n^N (h_i^n - \mu_i)(h_j^n - \mu_j) \tag{1}$$

where $N$ is the number of samples in the batch, $h$ are the feature activations at the selected layer, and $\mu$ is the sample mean of feature activations for the batch. Armed with this definition, they augment the standard cross-entropy loss function with the following regularization term:

$$\mathcal{L}_{DeCov} = \frac{1}{2}(||C||_F^2 - ||diag(C)||^2) \tag{2}$$

where $|| \cdot ||_F$ is the Frobenius norm. This term reduces the magnitude of the off-diagonal terms of the observed batch feature covariance matrix regardless of class.

**OrthoReg**  In (Rodríguez et al., 2017), the authors design a weight update technique that directly controls the cosine similarity between weight vectors throughout the neural network. The regularization cost function introduced is,

$$C(\theta) = \frac{1}{2} \sum_{i=1}^n \sum_{j=1,j\neq i}^n \left( \frac{\langle \theta_i, \theta_j \rangle}{||\theta_i||||\theta_j||} \right)^2 \tag{3}$$

where $\theta_i$ is the weight vector connecting to neuron $i$ of the next layer, which has n hidden units. Two different weight update rules are derived from this cost function: one that regularizes both positive and negatively correlated weights and another that penalizes only positive correlations. The authors hypothesize that negative correlations should not be penalized as they aide in generalization.

## 3  THEORETICAL FRAMEWORK

### 3.1  CHEBYSHEV'S PROTOTYPE RISK (CPR)

The concept we want to grasp is, if a class prototype feature vector is an ideal representation for it's class examples and if an example is classified correctly if it is similar in angular feature space to a prototype, then how can we compute or bound the probabilistic risk that an example is *dissimilar* to its class prototype, thus potentially resulting in a misclassification?

We first rename the cosine similarity function for two generic vectors:

$$CS(\boldsymbol{v}, \boldsymbol{u}) \stackrel{\text{def}}{=} \frac{\boldsymbol{v} \cdot \boldsymbol{u}}{||\boldsymbol{v}||||\boldsymbol{u}||} \tag{4}$$

We now define a similarity based measure that captures the average *dissimilarity* in feature space of all $K$ prototype feature vectors:

**Definition 1.** *Given a sufficiently trained classifier with low empirical risk, $f(\boldsymbol{x}, \theta)$, and set of $K$ prototype feature vectors of dimension $J$, $\{\boldsymbol{p}_1, ..., \boldsymbol{p}_K\}$, each being an ideal representation of a corresponding class $k$, the prototype dissimilarity value $DS \in [0, 1]$ is:*

$$DS \stackrel{def}{=} 1 - \frac{1}{K(K-1)} \sum_{i \neq j}^K CS(\boldsymbol{p}_i, \boldsymbol{p}_j) \tag{5}$$

Our lemma uses Chebyshev's inequality to bound the probability that an example deviates more than the prototype dissimilarity value from its prototype.

**Lemma 3.1.** *Given a sufficiently trained classifier with low empirical risk, $f(\boldsymbol{x}, \theta)$, a prototype feature vector $\boldsymbol{p}$ of dimension $J$, which is an ideal representation of a corresponding class $k$, a prototype dissimilarity value $DS$, a feature vector $\boldsymbol{v}$ for a class $k$ input example, and a covariance function $cov(X, Y) = \mathbb{E}[(X - \mathbb{E}[X])(Y - \mathbb{E}[Y])]$ for random variables $X$ and $Y$, the following inequality holds:*

$$Pr\big[|CS(\boldsymbol{v}, \boldsymbol{p}) - \mathbb{E}[CS(\boldsymbol{v}, \boldsymbol{p})]| \geq DS\big] \leq \frac{\sum\limits^{J} cov(\hat{v}_i\hat{p}_i, \hat{v}_j\hat{p}_j)}{DS^2} \tag{6}$$

*where $\hat{v}_j$ and $\hat{p}_j$ are the components of the unit feature vectors $\hat{\boldsymbol{v}}$ and $\hat{\boldsymbol{p}}$.*

The inequality in the above lemma is the two-tailed Chebyshev's inequality (Ross, 2007). If we assume that the expected cosine similarity between $\boldsymbol{v}$ and $\boldsymbol{p}$ is close to one, we can make use of Chebyshev-Cantelli's one-tailed version of the inequality (Boucheron et al., 2013):

**Corollary 3.1.1.** *If Lemma 3.1 holds and* $\mathbb{E}\big[CS(\boldsymbol{v}, \boldsymbol{p})\big] = 1.0$*, then the following inequality holds:*

$$Pr\big[CS(\boldsymbol{v}, \boldsymbol{p}) - E\big[CS(\boldsymbol{v}, \boldsymbol{p})\big] \leq -DS\big] \leq \frac{\sum\limits^{J} cov(\hat{v}_i \hat{p}_i, \hat{v}_j \hat{p}_j)}{\sum\limits^{J} cov(\hat{v}_i \hat{p}_i, \hat{v}_j \hat{p}_j) + DS^2} \tag{7}$$

*where* $\hat{v}_j$ *and* $\hat{p}_j$ *are the components of the unit feature vectors* $\hat{\boldsymbol{v}}$ *and* $\hat{\boldsymbol{p}}$*.*

For a full proof of lemma 3.1 and corollary 3.1.1, see Appendix A.1.

Although the above lemma and corollary only bound the probability that an example deviates from its prototype, they inform us on what quantities are important for minimizing the bounds on an example deviating from its prototype and risk being misclassified.

Quantitatively, the significance of the value $DS$ on the left-hand side is that once the example feature vector has deviated this amount in similarity from a source prototype, it is, on average, just as similar to all other prototypes than it was to the original - a situation that is precarious for classification.

The right hand side reinforces the canonical idea also held in Cogswell et al. (2016) that minimizing the covariance of learned features benefits the quality of the trained model in reducing overfitting. Additionally, the global prototype dissimilarity appears as the *square* of the mean dissimilarity, implying that prototypes that are very similar to each other likely pose the highest risk for misclassification.

There also exist numerical differences between the two-tailed and one-tailed versions of the inequality. Since $DS \in [0, 1]$, the two-tailed version of the inequality is in theory not finite for $DS = 0$, which is not the case in the single-tailed version if we assume very little ($\sum^{J} cov(\hat{v}_i \hat{p}_i, \hat{v}_j \hat{p}_j)$ is non-zero)). Even though $\mathbb{E}\big[CS(\boldsymbol{v}, \boldsymbol{p})\big] \neq 1$, it may be reasonable to assume that it is and empirically evaluate if the inequality correlates well with model quality, implying that the bound is useful.

Together, we consider these quantities to constitute a *Chebyshev Prototype Risk* (CPR) metric that is in the interest of the classifier to minimize during training and is defined as:

**Definition 2.** *Given an example feature vector* $\boldsymbol{v}$ *with true label* $y = k$ *and a prototypical (ideal) class* $k$ *feature vector* $\boldsymbol{p}_k$*, let the Chebyshev Prototype Risk (CPR) be defined by:*

$$Chebyshev\ Prototype\ Risk\ (CPR) \stackrel{def}{=} \frac{\sum\limits^{J} cov(\hat{v}_i \hat{p}_i, \hat{v}_j \hat{p}_j)}{DS^2} \tag{8}$$

Our probabilistic model for CPR has parallels to Neural Collapse property (NC1), which is the collapse of within-class feature covariance when a neural network has been trained for a sufficient number of epochs beyond 100% training accuracy (see (Papyan et al., 2020) for details). Through this lens, our work shares two connections to Neural Collapse. First, our results show that our algorithm significantly reduces intra-class feature covariance compared to baseline models within a typical duration (100 epochs) of training, which is not in the terminal phase of training – in this way our loss components accelerate the covariance effects of neural collapse, which has been shown to improve generalization in many settings and lends credence to our Chebyshev model that reducing CPR reduces overfitting (Papyan et al., 2020). The second connection is that we introduce a new probabilistic modeling of the similarity between an example's features to its class mean feature vector (Lemma 3.2), which to our knowledge has not been mathematically derived in the existing Neural Collapse discussion. Future work could look to integrate theoretical arguments like ours into modeling Neural Collapse phenomena.

## 3.2 Properties of Prototypes

We have thus far referred to the prototype feature vectors as "ideal" representations of their categories, one prototype for one class. In practice, we will employ a traditional idea in machine learning that a category's examples should be close together in feature space and each category would then have a representational centroid. In an online training setting where the prototypes, "centers", of each category can be updated while training, we define the feature space loss function for an example $(\boldsymbol{x}_n, y_n)$:

$$\mathcal{L}_{proto,n} = \sum_{k}^{K} \mathbb{1}(k = y_n) \, ||g(\boldsymbol{x}_n) - \boldsymbol{p}_k||^2 \tag{9}$$

If we assume convergence of the above loss component (Bottou et al., 2016), we can derive two important results regarding a prototype and its category's examples.

**Lemma 3.2.** *For a sufficiently trained network with low empirical risk on $\mathcal{L}_{proto}$, the features of each resulting prototype feature vector, $\boldsymbol{p}_k$ for class $k$ of dimension $J$, converge to $\boldsymbol{p}_{k,j} = \frac{1}{N_k}\sum_{n=1}^{N_k} g(\boldsymbol{x}_{n,k})_j$, where $N_k$ is the number of training examples from class $k$.*

In no uncertain terms, lemma 3.2 states that if we optimize the loss component $\mathcal{L}_{proto}$ and reach convergence, the resulting values of the prototype vectors in feature space will be the arithmetic means of the activations of each prototype's respective training examples' feature vector values.

Optimizing the $\mathcal{L}_{proto}$ also equates to minimizing the squared residuals of each feature component between class examples and class prototypes. But if the prototypes converge to the class feature means, then we can show an additional important aspect of prototype convergence:

**Corollary 3.2.1.** *If Lemma 3.2 holds, then minimizing $\mathcal{L}_{proto}$ is equivalent to minimizing the individual feature variances over the training examples, $\sum_{j=1}^{J} \mathbb{V}(g(\boldsymbol{x}_{n,k})_j)$ for each class $k$.*

Corollary 3.2.1 formally states that if the prototypes represent the sample mean feature values of categories and we minimize the squared residuals between examples and prototypes, then we are trying to minimize the sample variance of each class' features. For a full proof of lemma 3.2 and corollary 3.2.1, see Appendix A.2.

## 4 Approach and Algorithm

We contribute a training algorithm that explicitly and efficiently minimizes CPR. Our training algorithm is exemplified by a multi-component loss function that is optimized during training. For each drawn mini-batch of examples during training, our computed loss function is:

$$\mathcal{L} = \underbrace{\mathcal{L}_{CE}}_{\text{cross-entropy}} + \underbrace{\beta\mathcal{L}_{proto}}_{\text{Eqn. 9}} + \underbrace{\gamma\mathcal{L}_{cov}}_{\boldsymbol{vp}\text{-covariance}} + \underbrace{\zeta\mathcal{L}_{CS}}_{\boldsymbol{p}\text{-similarity}} \tag{10}$$

where $\mathcal{L}_{CE}$ is the cross entropy loss, $\mathcal{L}_{proto}$ is the example-prototype loss in Eq. 9, $\mathcal{L}_{cov}$ is a covariance loss, and $\mathcal{L}_{CS}$ is a loss on global prototype cosine similarity. The hyperparameters $\beta$, $\gamma$, and $\zeta$ are the relative weights of the appropriate loss components.

**Computation of $\mathcal{L}_{CS}$** Per the derived inequalities in lemma 3.1 and corollary 3.1.1, we would need to maximize the global prototype dissimilarity in order decrease the probability bound. In practice, we prefer the training dynamics of minimizing the prototype similarities such that the loss has a lower bound of zero. Further, the prototype similarity dependency is quadratic, implying that our loss should also have a quadratic form. Given the current state of the prototype set $\{\boldsymbol{p}_1, ..., \boldsymbol{p}_K\}$, we calculate $\mathcal{L}_{CS}$ as:

$$\mathcal{L}_{CS} = \frac{1}{K(K-1)} \sum_{i \neq j}^{K} (CS(\boldsymbol{p}_i, \boldsymbol{p}_j))^2 \tag{11}$$

In words, we compute the squared cosine similarity between all-pairs of prototype vectors, excluding the diagonal, at every minibatch and attempt to minimize the mean. By squaring the loss term, the gradients tend to focus on the most similar pairings of prototypes.

**Computation of $\mathcal{L}_{cov}$** The computation of the covariance function for class $k$, $cov(\hat{v}_i\hat{p}_i, \hat{v}_j\hat{p}_j)_k = E[(\hat{v}_i\hat{p}_i - E[\hat{v}_i\hat{p}_i])(\hat{v}_j\hat{p}_j - E[\hat{v}_j\hat{p}_j])]_k$, is simplified by freezing the state of the prototype for the computation of this loss component such that it acts as a vector of constants for $\mathcal{L}_{cov}$ only. We drop the subscript $k$ for brevity, but emphasize that this loss component is computed only between an example's features and the prototype features of its true class. When we draw an input example during training, $\hat{p}$ is selected according to the example's label. The simplified form is,

$$\begin{aligned}
cov(\hat{v}_i\hat{p}_i, \hat{v}_j\hat{p}_j) &= \mathbb{E}[(\hat{v}_i\hat{p}_i - \hat{p}_i\mathbb{E}[\hat{v}_i])(\hat{v}_j\hat{p}_j - \hat{p}_j\mathbb{E}[\hat{v}_j])] \\
&= \mathbb{E}[\hat{p}_i(\hat{v}_i - \mathbb{E}[\hat{v}_i])\hat{p}_j(\hat{v}_j - \mathbb{E}[\hat{v}_j])] \\
&= \mathbb{E}[\hat{p}_i(\hat{v}_i - \hat{p}_i)\hat{p}_j(\hat{v}_j - \hat{p}_j)]
\end{aligned} \tag{12}$$

Algorithm 1 (see Appendix Sec. A.6 for pseudocode) provides an efficient implementation to minimize the intra-class feature covariance terms in Eqn. 12. Utilizing lemma 3.2, we treat the learned class prototypes as the running arithmetic means of each class' examples and therefore as we optimize $\mathcal{L}_{proto}$, the prototypes constantly adjust the mean of their class examples. In mathematical terms, we can replace the term $E[\hat{v}_i]$ with the current value of the appropriate prototype.

Instead of calculating the all-pairs feature covariance matrix in $\mathcal{O}(J^2)$ time, we compute an effective approximation in $\mathcal{O}(J + JlogJ)$. We first identify the correct prototype feature vector for each example based on its label, sort the features of the selected prototypes, reindex the examples' features by the sorted indices of their prototype, compute their activation differences, randomly re-align (shift) these differences by padding with zeros, and then compute their padded element-wise product.

By shifting the example and prototype features relative to each other, we ensure that $i \neq j$ in Eqn. 12 and by sorting per the prototype values, we allow the highest (most important) features of the class to be compared. The user-parameter $\nu$ allows the loss term to regularize positive, negative, or both possible signs of covariance terms.

**Loss Summary** The total loss function in Eq. 10 works as a symbiotic system, each contributing the following:

- $\mathcal{L}_{CE}$ fits the classifier decision boundary to the training examples.
- $\mathcal{L}_{proto}$ maintains the class prototypes as the mean feature vectors of their respective classes by lemma 3.2 and thus makes the prototypes useful in the covariance calculations of $\mathcal{L}_{cov}$. By corollary 3.2.1, this loss function also minimizes the diagonal terms of the intra-class covariance matrices.
- $\mathcal{L}_{cov}$ regularizes the off-diagonal terms of the intra-class feature covariance matrices.
- $\mathcal{L}_{CS}$ reduces the global similarity between class prototypes.

## 5 EMPIRICAL EVALUATION

Many previous efforts evaluate new techniques by training on the full available training set with different model initializations and report a test accuracy containing variation over model initialization. We suggest that a robust way to assess overfitting tendency is to

randomly draw many different training subsets from the full available training set because we need to test whether the learning algorithm resists overfitting *regardless of the seen examples*. For our assessments, we randomly draw 12 training subsets from the available training data, each being 50% of the size of the source set. We draw these training subsets in a stratified manner such that each category has the same number of samples. We note that this evaluation method is the root idea behind bias-variance decomposition (Yang et al., 2020).

**Datasets, Architectures, Training** We applied our algorithm and previous works to the well-known image classification data sets CIFAR100 (Krizhevsky et al.) and STL10 (Coates et al., 2011). Both datasets make available 500 examples per class category (from which we randomly draw 250 every time we instantiate a training subset) and consist of 3-channel color images of natural objects. The main differences are in number of categories, 100 for CIFAR100 and 10 for STL10, and input image size, 32x32 for CIFAR100 and 96x96 for STL10. We trained Residual Network (He et al., 2016) based architectures for both datasets, using a ResNet18 at 50% width for CIFAR100 and ResNet34 at 50% width for STL10. We maintained the default depths for both networks, but reduced the width to expedite computations. We trained all models for 100 epochs using stochastic gradient descent (momentum=0.9) on a cosine annealed learning schedule beginning at a learning rate of 0.1, used a batch size of 128, and varied the weight decay depending on the case being studied (see result tables). All runs were computed on a single GPU. We provided a warmup period to all trainings of 10 epochs before any regularizers were applied. Standard flipping and cropping were employed for data augmentation in all runs.

We used the default settings for OrthoReg in both datasets. For DeCov, we used the suggested loss hyperparameter of 0.1 for CIFAR100, but had trouble fitting the STL10 data at that value. We reduced the DeCov loss hyperparameter to 0.01 for STL10 in order to fit the data. For our algorithm, we could not conduct a large hyperparameter study on $(\beta,\gamma,\zeta)$ for computational reasons, so we instead performed cross-entropy loss training on the first training subset for 10 epochs and adjusted the values of $(\beta,\gamma,\zeta)$ until the other 3 loss components $\mathcal{L}_{cov}$, $\mathcal{L}_{CS}$, and $\mathcal{L}_{proto}$ had similar magnitude to $\mathcal{L}_{CE}$ at epoch 10. Our goal was to ensure that after the 10 epoch warmup period the relative scales of the different losses were similar and that the full training could optimize the Chebyshev Prototype Risk (CPR) while still fitting the data. Our results indicate that our hyperparameter choices indeed allowed the model to fit the data to 100% training accuracy while still reducing the CPR.

**Results** We ran all regularization algorithms for two choices of weight decay (0.0,5e-4) on CIFAR100 and (5e-4) on STL10 based on the previous experiences of (Cogswell et al., 2016) and (Rodríguez et al., 2017) that weight-decay achieves good regularization on its own, thus washing out some of the effect of additional regularization terms. For both weight decay settings, we ran a baseline consisting of only the cross-entropy loss component for comparison. Tables 1 and 2 show that our algorithm is effective in boosting generalization performance on CIFAR100 regardless of the training set selected when compared to either the baseline, OrthoReg, or DeCov. Our algorithm's effect is more noticeable in the 0.0 weight decay setting as expected, but still maintains an improvement in performance in the 5e-4 weight decay environment.

Additionally, we ran our algorithm for the three possible selections of $\nu$. The results were very similar for whether we controlled only positive covariance $\nu = 1$ or both positive and negative ($\nu = 0$) so we report only the $\nu = 0$ case. In the 0.0 weight decay setting, there is no significant difference in changing $\nu$, but for 5e-4 weight decay, regularizing the negative covariance terms provides a marginal reduction in overfitting. We saw similar trends for training a ResNet34 on the STL10 - for details refer to Appendix A.4.

Figure 2 shows the average Chebyshev Risk metric of lemma 3.1 across all CIFAR100 classes. Clearly, reducing the risk metric *can* lead to models that generalize better on unseen examples, but we also note that there are models with low risk that have only moderate generalization compared to the baseline. The trends suggest that minimizing CPR is necessary to maximize the generalization of a selected architecture on a given dataset, but it is

Table 1: CIFAR100, ResNet18 Test Accuracy for Weight Decay = 0.0

| | Test Accuracy | | | | |
| | Baseline | Decov | OrthoReg | Ours | Ours |
| Train Set | | w=0.1 | | $\nu = 0$ | $\nu = -1$ |
| 1 | 0.591 | 0.602 | 0.585 | **0.628** | 0.616 |
| 2 | 0.589 | 0.614 | 0.598 | **0.629** | 0.626 |
| 3 | 0.601 | 0.598 | 0.593 | 0.617 | **0.621** |
| 4 | 0.596 | 0.590 | 0.608 | **0.622** | **0.622** |
| 5 | 0.595 | 0.605 | 0.598 | **0.624** | 0.615 |
| 6 | 0.596 | 0.612 | 0.598 | **0.624** | 0.615 |
| 7 | 0.592 | 0.612 | 0.582 | 0.618 | **0.628** |
| 8 | 0.596 | 0.618 | 0.595 | 0.617 | **0.624** |
| 9 | 0.595 | 0.609 | 0.592 | 0.613 | 0.619 |
| 10 | 0.581 | 0.597 | 0.596 | 0.621 | **0.626** |
| 11 | 0.600 | 0.612 | 0.606 | 0.621 | **0.630** |
| 12 | 0.597 | 0.619 | 0.594 | **0.635** | 0.621 |
| Mean | 0.594 | 0.607 | 0.595 | **0.622** | **0.622** |
| $\sigma$ | 0.005 | 0.009 | 0.008 | 0.006 | 0.005 |
| Min | 0.581 | 0.590 | 0.582 | 0.613 | **0.615** |

Table 2: CIFAR100, ResNet18 Test Accuracy for Weight Decay = 0.0005

| | Test Accuracy | | | | |
| | Baseline | Decov | OrthoReg | Ours | Ours |
| Train Set | | w=0.1 | | $\nu = 0$ | $\nu = -1$ |
| 1 | 0.662 | 0.669 | 0.663 | 0.667 | **0.673** |
| 2 | 0.659 | **0.672** | 0.664 | 0.669 | **0.672** |
| 3 | 0.664 | 0.665 | 0.657 | 0.663 | **0.671** |
| 4 | 0.662 | 0.660 | 0.665 | 0.671 | **0.673** |
| 5 | 0.666 | 0.664 | 0.662 | 0.660 | **0.676** |
| 6 | 0.665 | 0.668 | 0.670 | 0.670 | **0.677** |
| 7 | 0.662 | 0.667 | 0.664 | 0.670 | **0.672** |
| 8 | 0.660 | 0.669 | 0.660 | 0.668 | **0.670** |
| 9 | 0.660 | 0.667 | 0.664 | 0.662 | **0.673** |
| 10 | 0.660 | 0.666 | 0.660 | 0.664 | **0.670** |
| 11 | 0.660 | 0.673 | 0.659 | **0.675** | 0.673 |
| 12 | 0.658 | **0.668** | 0.655 | **0.668** | **0.668** |
| Mean | 0.661 | 0.667 | 0.662 | 0.667 | **0.672** |
| $\sigma$ | 0.002 | 0.003 | 0.004 | 0.004 | 0.003 |
| Min | 0.658 | 0.660 | 0.655 | 0.660 | **0.668** |

not sufficient to guarantee it. To see the equivalent chart for the Chebyshev-Cantelli CPR, see Appendix A.5.

**Discussion** There is a rich literature on regularization algorithms that optimize feature decorrelation or separation in one or more layers of a model such as (Hui et al., 2023; Deng et al., 2022; Ayinde et al., 2019; Huang et al., 2018; Pereyra et al., 2017; Liu et al., 2016; Wan et al., 2013; Hinton et al., 2012). Almost all of these methods work off the notion that feature decorrelation and separability are beneficial to the test performance of a network. Our objective is to unify these approaches under a common mathematical model that details why these behaviors are desired. By doing this, our CPR metric provides the relative numerical importance between intra-class feature covariance (weighted by prototype activations) and inter-class separation (quadratic in prototype dissimilarity). While we wanted to show that explicitly reducing CPR in our algorithm improves overfitting over an unregularized baseline, we critically wanted to show that other non-CPR regularization techniques still implicitly improved CPR and resulted in better test performance, which would confirm our probabilistic model's relationship to misclassification (reducing the CPR for each class reduces misclassification risk). We chose OrthoReg and DeCov because they are simple, effective, and address feature decorrelation differently by either the weights

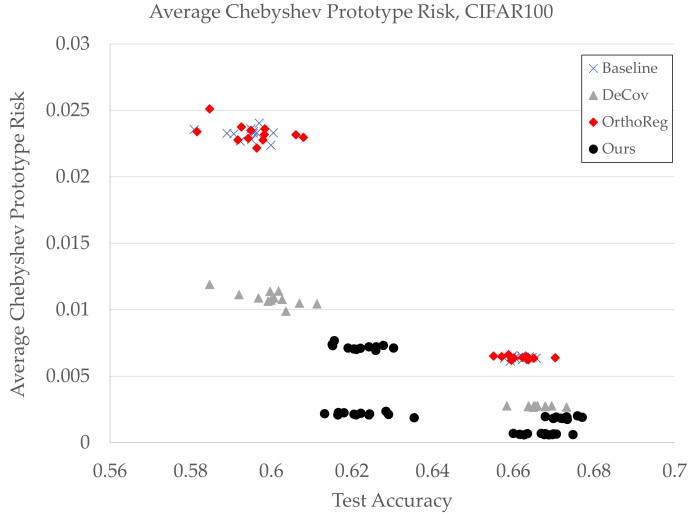

Figure 2: Assessment of CPR all training subsets and algorithms.

(OrthoReg) or the activations (DeCov). It would be worthwhile to evaluate all the methods in a similar training regime and compare their CPR metric.

We highlight some important differences and advantages of our method. First, our probability model informs us on the specific mathematical forms of intra-class covariance vs. class separation. Our feature covariance is uniquely prototype-weighted since we use $\mathbb{E}[p_i(v_i - p_i)p_j(v_j - p_j)]$. All categories have differently shaped prototype feature vectors and thus our weighted covariance will scale each feature gradient differently in backpropagation depending on its relative importance to each class. We further wanted to design a covariance algorithm that reduces the computational complexity. Using our feature sorting and padding algorithm, our method computes the covariance contributions for a sample in $\mathcal{O}(J + JlogJ)$ time, in comparison to previous approaches in $\mathcal{O}(J^2)$ time. Our results show that over the course of training, this method still reduces model feature covariance even though we do not explicitly compute the full covariance matrix. This gives our algorithm a distinct scaling advantage.

### 5.1 CONCLUSION

Many efforts for overfitting reduction to improve test performance have been shown effective. It has been shown that overfitting can be reduced by amending the standard cross-entropy loss with terms in one or more hidden layers of the network, including the convolutional layers. To build upon the wisdom guiding these previous works, we analytically tried to understand how class feature relationships affect misclassification.

Our analysis began by assuming a DNN is the composition of a feature extractor and classifier, where the classifier is the last fully connected layer of the network and the feature layer is the input vector to the classifier. Assuming that, corresponding to each class, there exists an ideal feature vector which we designate as a class prototype.

Formally, we derived Chebyshev and Chebyshev-Cantelli probability bounds on the deviation of cosine similarity between the features of an example and its class prototype. We added the terms in the inequality to define our novel loss function for optimizing the feature layer, but the new loss function backpropagates errors to the previous convolutional layers' and optimizes their parameter values as well. The new loss function based on our probability bounds effectively reduces intra-class feature covariance while keeping class examples separated in feature space, which reduces the risk of overfitting. Empirical results on multiple datasets and network architectures validates that the our loss function reduces overfitting and improves upon previous approaches in an efficient manner.

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

## A   APPENDIX

You may include other additional sections here.

### A.1   PROOFS FOR SECTION 3.1

The following is a proof of lemma 3.1 and corollary 3.1.1:

*Proof.* Given some non-negative random variable $X$, we begin by analyzing the continuous definition of expectation and derivation of Markov's inequality (Ross, 2007),

$$\mathbb{E}[X] = \int_{-\infty}^{\infty} x f(x) dx \tag{13}$$

In addition to $X$ being non-negative, in our application, we will being using cosine similarity, so are random variable is also bound above by 1:

$$\mathbb{E}[X] = \int_0^1 x f(x) dx \tag{14}$$

We can further split the integral by some value $a$,

$$\mathbb{E}[X] = \int_0^a x f(x) dx + \int_a^1 x f(x) dx \tag{15}$$

We can then chain together several inequalities,

$$\begin{aligned}
\mathbb{E}[X] &= \int_0^a x f(x) dx + \int_a^1 x f(x) dx \\
&\geq \int_a^1 x f(x) dx \\
&\geq \int_a^1 a f(x) dx \\
&= a \int_a^1 f(x) dx \\
&= a Pr[X \geq a]
\end{aligned} \tag{16}$$

This results in the final form for Markov's inequality,

$$Pr[X \geq a] \leq \frac{\mathbb{E}[X]}{a} \tag{17}$$

Let the non-negative random variable $X$ be $(X - E[X])^2$ and consider some value for $a$ on the interval $[0, 1]$, then Markov's inequality becomes,

$$\begin{aligned}
Pr[(X - E[X])^2 \geq a^2] &\leq \frac{\mathbb{V}[X]}{a^2} \\
Pr[|X - E[X]| \geq a] &\leq \frac{\mathbb{V}[X]}{a^2}
\end{aligned} \tag{18}$$

Given a prototype feature vector $\boldsymbol{p}$ for class $k$ and the feature vector of an example from class $k$, $\boldsymbol{v} = g(\boldsymbol{x})$, we consider the random variable $X$ to be the cosine similarity between the example's feature vector and it's class prototype:

$$X = \frac{\boldsymbol{v} \cdot \boldsymbol{p}}{||\boldsymbol{v}|| ||\boldsymbol{p}||} \stackrel{\text{def}}{=} CS(\boldsymbol{v}, \boldsymbol{p}) \tag{19}$$

We note that the cosine similarity in this case is a summation of dependent random variables,

$$X = \frac{\sum_j^J v_j p_j}{||\boldsymbol{v}|| ||\boldsymbol{p}||} \tag{20}$$

Furthermore, we can think of the summation as the sampling of two unit vectors $\hat{\boldsymbol{v}}$ and $\hat{\boldsymbol{p}}$ from the unit circle so that X becomes the summation of unit vector components, each component being a random variable:

$$X = \sum_{j}^{J} \hat{v}_j \hat{p}_j \tag{21}$$

Substituting into Eq. 18, we convert the inequality to bound the probability that the similarity between a class sample's unit feature vector deviates more than a value $a$ from its expectation:

$$Pr\big[\big|CS(\boldsymbol{v},\boldsymbol{p}) - E\big[CS(\boldsymbol{v},\boldsymbol{p})\big]\big| \geq a\big] \leq \frac{\sum\limits^{J} \mathrm{cov}(\hat{v}_i\hat{p}_i, \hat{v}_j\hat{p}_j)}{a^2} \tag{22}$$

Intuitively, as the similarity between an examples unit feature vector and a prototype unit vector from class $k$ decreases, the chance of that example being classified into $k$ decreases. For a given prototype $\boldsymbol{p}$, there also exist $(k-1)$ dissimilar prototypes towards which an example could become more similar as it deviates from the original prototype.

We define the mean dissimilarity between prototypes as follows and select it as a meaningful value for $a$ to represent the mean spacing between prototypes in feature space:

$$DS \stackrel{\mathrm{def}}{=} 1 - \frac{1}{K(K-1)} \sum_{i\neq j}^{K} CS(\boldsymbol{p}_i, \boldsymbol{p}_j) \tag{23}$$

Substituting into Eq. 22, we reach our final form for Chebyshev's two-side inequality:

$$Pr\big[\big|CS(\boldsymbol{v},\boldsymbol{p}) - E\big[CS(\boldsymbol{v},\boldsymbol{p})\big]\big| \geq DS\big] \leq \frac{\sum\limits^{J} \mathrm{cov}(\hat{v}_i\hat{p}_i, \hat{v}_j\hat{p}_j)}{DS^2} \tag{24}$$

We then deduce a one-sided version of Eq. 24 known as Chebyshev-Cantelli's inequality. If we assume that $E\big[CS(\boldsymbol{v},\boldsymbol{p})\big] \approx 1$, then the one-sided inequality is applicable. We refer to (Ross, 2007; Boucheron et al., 2013) for the steps required to make the deduction after we make this assumption. The final one-sided form is,

$$Pr\big[CS(\boldsymbol{v},\boldsymbol{p}) - E\big[CS(\boldsymbol{v},\boldsymbol{p})\big] \leq -DS\big] \leq \frac{\sum\limits^{J} \mathrm{cov}(\hat{v}_i\hat{p}_i, \hat{v}_j\hat{p}_j)}{\sum\limits^{J} \mathrm{cov}(\hat{v}_i\hat{p}_i, \hat{v}_j\hat{p}_j) + DS^2} \tag{25}$$

Thus completes our proof. □

## A.2 Proofs for Section 3.2

The following is a proof of lemma 3.2 and corollary 3.2.1:

*Proof.* We start by restating a theorem and corollary from (Bottou et al., 2016):

**Theorem A.1.** *(Bottou et al., 2016)(Nonconvex Objective, Diminishing Stepsizes). Under Assumptions 4.1 and 4.3, suppose that the SG method (Algorithm 4.1) is run with a stepsize sequence satisfying (4.19). Then with $A_Z \stackrel{def}{=} \sum_{z=1}^{Z} \alpha_z$,*

$$\lim_{Z\to\infty} \mathbb{E}\big[ \sum_{z=1}^{Z} \alpha_z ||\nabla F(w_z)||^2 \big] < \infty$$

$$\textit{and therefore } \mathbb{E}\big[ \frac{1}{A_Z} \sum_{z=1}^{Z} \alpha_z ||\nabla F(w_z)||^2 \big] \xrightarrow{Z\to\infty} 0 \tag{26}$$

In this theorem, $z$ is the iteration number of the stochastic gradient descent method ("SG"), $F$ is the objective function, and $w$ are the model parameters. For full details, see (Bottou et al., 2016).

As a corollary to Theorem A.1,

**Corollary A.1.1.** *(Bottou et al., 2016) Under the conditions of Theorem A.1, if we further assume that the objective function $F$ is twice differentiable, and that the mapping $w \mapsto ||\nabla F(w_z)||^2$ has Lipschitz-continuous derivatives, then*

$$\lim_{Z \to \infty} \mathbb{E}\left[||\nabla F(w_z)||^2\right] = 0 \tag{27}$$

We assume that corollary A.1.1 is true for our prototype objective function, which we restate here for a single example $(\boldsymbol{x}_n, y_n)$ and $\boldsymbol{p}_k \in \mathbb{R}^J$:

$$\mathcal{L}_{proto,n} = \sum_{k}^{K} \mathbb{1}(k = y_n) \, ||g(\boldsymbol{x}_n) - \boldsymbol{p}_k)||^2 \tag{28}$$

At each iteration $z$, we randomly select a sample from the training set and compute the losses and gradients produced from the sample for the current update. The stochastic gradient descent update rule is:

$$\boldsymbol{p}_{k,z+1} \leftarrow \boldsymbol{p}_{k,z} - \alpha_z \nabla_{\boldsymbol{p}_k} \mathcal{L}_{proto,n} \tag{29}$$

We can expect the gradient update rule on $\boldsymbol{p}$ to converge such that:

$$||\frac{1}{N} \sum_{n=1}^{N} \nabla_{\boldsymbol{p}_k} \mathcal{L}_{proto,n}|| = 0 \tag{30}$$

We can expand Eqn. 30 using Eqn. 28 and then re-write $||g(\boldsymbol{x}_n) - \boldsymbol{p}_k)||^2$ as a summation over its $J$ elements:

$$||\nabla_{\boldsymbol{p}_k} \frac{1}{N} \sum_{n=1}^{N} \sum_{j=1}^{J} \mathbb{1}(k = y_n) \, (g(\boldsymbol{x}_n)_j - p_{k,j})^2|| = 0 \tag{31}$$

The outer norm, $||\cdot||$, can represent a summation over each individual gradient term produced by $\nabla_{\boldsymbol{p}_k}$. We replace the indicator function by subscripting $\boldsymbol{x}$ with $k$ to indicate it has true label $k$. Knowing this, we can introduce a summation over the $J$ elements of $\nabla_{\boldsymbol{p}_k}$, which we represent by its individual partial derivatives, $\frac{\partial}{\partial p_j}$ and take the square of both sides to produce,

$$\sum_{j=1}^{J} \left( \frac{\partial}{\partial p_{k,j}} \frac{1}{N_k} \sum_{n=1}^{N_k} \sum_{j=1}^{J} (g(\boldsymbol{x}_{n,k})_j - p_{k,j})^2 \right)^2 = 0 \tag{32}$$

Applying each partial derivative, we get,

$$\sum_{j=1}^{J} \left( \frac{1}{N_k} \sum_{n=1}^{N_k} \sum_{j=1}^{J} 2(g(\boldsymbol{x}_{n,k})_j - p_{k,j}) \right)^2 = 0 \tag{33}$$

We observe that equations $\mathbb{1}(k = y_n) \frac{1}{N_k} \sum_{n=1}^{N_k} 2(g(\boldsymbol{x}_{n,k})_j - p_{k,j}) = 0, \quad j = 1...J$ are a solution to Eqn. 33 and then find an expression for $g(\boldsymbol{x}_{n,k})_j$:

$$-p_{k,j} + \frac{1}{N_k} \sum_{i=1}^{N_k} g(\boldsymbol{x}_{n,k})_j = 0 \quad j = 1...J$$

$$p_{k,j} = \frac{1}{N_k} \sum_{i=1}^{N_k} g(\boldsymbol{x}_{n,k})_j \quad j = 1...J \tag{34}$$

$$p_{k,j}^* = \frac{1}{N_k} \sum_{i=1}^{N_k} g(\boldsymbol{x}_{n,k})_j$$

We assume that we apply minimize the empirical risk of $\mathcal{L}_{proto,n,k}$ such that Eqn. 34 applies and then we restate $\mathcal{L}_{proto,n,k}$:

$$\frac{1}{N_k} \sum_{n=1}^{N_k} \mathcal{L}_{proto,n,k} = \frac{1}{N_k} \sum_{n=1}^{N_k} ||g(\boldsymbol{x}_{n,k}) - \boldsymbol{p}_k||^2 \tag{35}$$

We write $||g(\boldsymbol{x}_{n,k}) - \boldsymbol{p}_k||^2$ as a summation over its $J$ elements and substitute in Eqn. 34 for $\boldsymbol{p}_k$ to find a concise expression for the empirical risk of $\mathcal{L}_{proto}$.

$$\frac{1}{N_k} \sum_{n=1}^{N_k} \mathcal{L}_{proto,n,k} = \sum_{j=1}^{J} \frac{1}{N_k} \sum_{n=1}^{N_k} (g(\boldsymbol{x}_{n,k})_j - p_{k,j})^2$$

$$= \sum_{j=1}^{J} \frac{1}{N_k} \sum_{n=1}^{N_k} \left( g(\boldsymbol{x}_{n,k})_j - \frac{1}{N_k} \sum_{n=1}^{N_k} g(\boldsymbol{x}_{n,k})_j \right)^2 \tag{36}$$

$$= \sum_{j=1}^{J} \mathbb{V}(g(\boldsymbol{x}_{n,k})_j) \quad k = 1...K$$

where $\mathbb{V}(g(\boldsymbol{x}_{n,k})_j)$ are the individual feature vector variances over the training examples.

$\square$

## A.3 Hyperparameters

- (CIFAR100, ResNet18, $\nu = 0$, $\beta = 70$, $\gamma = 25e3$, $\zeta = 1$)
- (CIFAR100, ResNet18, $\nu = -1$, $\beta = 70$, $\gamma = 50e3$, $\zeta = 1$)
- (STL10, ResNet34, $\nu = 0$, $\beta = 8$, $\gamma = 50e3$, $\zeta = 0.2$ )
- (STL10, ResNet34, $\nu = -1$, $\beta = 8$, $\gamma = 200e3$, $\zeta = 0.2$ )

## A.4 STL10 Results

Results for STL10 indicate that DeCov, OrthoReg, and our algorithm have very similar overfitting mitigation in the weight-decayed setting. All algorithms show improved performance over the weight-decay only baseline case. Overall we see much more variation in the STL10 data, which could originate from being more sensitive to model initialization or the size of the selected training sets was too small to provide consistent generalization.

Table 3: STL10, ResNet34 Test Accuracy for Weight Decay = 0.0005

| | Test Accuracy | | | | | |
| | Baseline | Decov | Decov | OrthoReg | Ours | Ours |
| Train Set | | w=0.01 | w=0.1 | | $\nu = 0$ | $\nu = -1$ |
| 1 | 0.571 | **0.685** | 0.578 | 0.624 | 0.635 | 0.677 |
| 2 | 0.692 | **0.687** | 0.563 | 0.645 | 0.677 | 0.660 |
| 3 | 0.579 | 0.621 | 0.444 | 0.630 | 0.650 | **0.670** |
| 4 | 0.626 | 0.658 | 0.620 | **0.679** | 0.631 | 0.661 |
| 5 | 0.664 | **0.665** | 0.638 | 0.614 | 0.644 | 0.654 |
| 6 | 0.653 | 0.673 | 0.584 | **0.694** | 0.684 | 0.660 |
| 7 | 0.629 | 0.656 | 0.629 | **0.661** | 0.644 | 0.647 |
| 8 | 0.670 | **0.685** | 0.665 | 0.628 | 0.663 | 0.632 |
| 9 | 0.625 | 0.675 | 0.584 | 0.555 | 0.680 | **0.684** |
| 10 | 0.608 | 0.644 | **0.657** | 0.649 | 0.583 | **0.657** |
| 11 | **0.685** | 0.665 | 0.585 | 0.654 | 0.669 | 0.666 |
| 12 | 0.680 | 0.651 | 0.629 | 0.670 | 0.659 | **0.684** |
| Mean | 0.640 | **0.664** | 0.598 | 0.642 | 0.652 | 0.662 |
| $\sigma$ | 0.040 | 0.019 | 0.059 | 0.036 | 0.028 | 0.015 |
| Min | 0.571 | 0.621 | 0.444 | 0.555 | 0.583 | **0.632** |

## A.5 Chebyshev-Cantelli Prototype Risk, CIFAR100

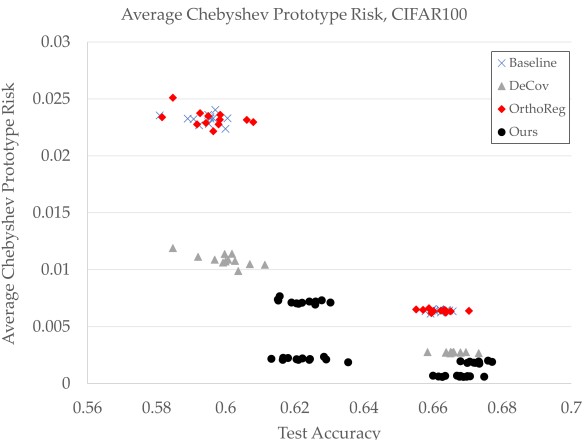

Figure 3: Assessment of Chebyshev-Cantelli Prototype Risk on right hand side of corollary 3.1.1 for all training subsets and algorithms.

## A.6 Pseudocode for Algorithm 1

---

**Algorithm 1:** Computation of $\mathcal{L}_{cov,n}$ for single example, class $k$

---

**Input:** $\boldsymbol{v}_k = g(\boldsymbol{x}_{n,k}; \theta_g) \in \mathbb{R}^J$, $\boldsymbol{p}_k \in \mathbb{R}^J$, $\nu$
**Output:** $\mathcal{L}_{cov,n}$
$\hat{\boldsymbol{v}_k} \leftarrow normalize(\boldsymbol{v}_k)$
$\hat{\boldsymbol{p}_k} \leftarrow normalize(\boldsymbol{p}_k)$
$\hat{\boldsymbol{p}_k} \leftarrow sort(\hat{\boldsymbol{p}_k})$
$\hat{\boldsymbol{v}_k} \leftarrow reindex(\hat{\boldsymbol{v}_k})$
`// rearrange` $\hat{\boldsymbol{v}_k}$ `by the sorted indices of` $\hat{\boldsymbol{p}_k}$
$\delta = \hat{\boldsymbol{p}_k} \odot (\hat{\boldsymbol{v}_k} - \hat{\boldsymbol{p}_k})$ `// same as` $\hat{p}_i(\hat{v}_i - \hat{p}_i)$
$r \leftarrow randint(1, 10)$  `// random integer from 1-10`
$\delta_{pad,L} \leftarrow PadLeftZeros(\delta, r)$  `// Pad` $r$ `zeros on left`
$\delta_{pad,R} \leftarrow PadRightZeros(\delta, r)$  `// Pad` $r$ `zeros on right`
**if** $\nu == 0$ **then**
$\quad|\quad Z \leftarrow |\delta_{pad,L} \odot \delta_{pad,R}|$
**else**
$\quad|\quad Z \leftarrow ReLU(sign(\nu)(\delta_{pad,L} \odot \delta_{pad,R}))$
**end**
return $\mathcal{L}_{cov,n} = \frac{1}{J+r} \sum_{j=1}^{J+r} Z$

---

