# OpenReview forum: "Minimizing Chebyshev Risk Magically Mitigates the Perils of Overfitting"
_ICLR.cc/2024/Conference — Submitted to ICLR 2024_

### Official Review · Reviewer_wkG8 · 2023-10-23

**Soundness:** 2 fair
**Presentation:** 2 fair
**Contribution:** 2 fair
**Rating:** 3
**Confidence:** 3

**Summary:**

This paper proposes a novel regularization for deep neural networks (DNNs) based on Chebyshev's inequality, where Chebyshev's inequality is used to derive the upper bound of the probability of an embedding feature for an example deviating from class-wise prototypes.
Losses for estimating prototypes as the class-wise embedding average, reducing intra-class feature covariances, and making prototypes orthogonal to each other are proposed.
Experiments are conducted to compare the proposed regularization with existing methods that try to minimize covariances between activations or weights.

**Strengths:**

- The use of Chebyshev's inequality to derive the regularization for DNNs is novel.

**Weaknesses:**

- I could not figure out the theoretical justification for using DS in Lemma 3.1 or Corollary 3.1.1.
    - If I understand correctly, the DS part in Eq.(6) can be any positive variable. Then what is the reason for using DS here?
    - Moreover, the authors claim to regularize DNN training by increasing DS (which is established by decreasing $\mathcal{L}_{CS}$), because it leads to a smaller value of the right part of Eq.(6). However, the larger DS value leads to the looser condition from the point of view of the left part of Eq.(6).
- Discussion and empirical comparison with related work is insufficient.
    - There are several other existing papers that discuss the orthogonality of weights, such as [1].
    - It is also preferable to qualitatively or qualitatively compare the proposed method with other methods using class-wise prototypes, such as [2].
    - Formatting in references is incomplete. For example, some papers do not have a place of publication.
- Experiments are performed with CIFAR-100 and STL-10 only.

[1] L. Huang et al., Orthogonal Weight Normalization: Solution to Optimization over Multiple Dependent Stiefel Manifolds in Deep Neural Networks, AAAI 2018.

[2] J. Deng et al., ArcFace: Additive Angular Margin Loss for Deep Face Recognition, CVPR 2019.

**Questions:**

- In Lemma 3.1, is there an assumption that the class label of $v$ is $k$?

- In Section 5.4, I could not understand how the hyperparameters are determined in the proposed method.

- In Section 4.3, Eq.11 -> Eq.9?

---

> ### Author Response · Authors · 2023-11-17
>
> Hello Reviewer wkG8,
>
> Thank you for reviewing our paper in-depth.  Your questions and comments are very detailed and helpful.  We will try to answer them one-by-one as best we can.
>
> In Lemma 3.1, is there an assumption that the class label of v is k?
> Yes.  We propose to change the wording of lemma 3.1 to read:
> “Lemma 4.3.1 Given a sufficiently trained classifier … , [a feature vector v for a class k input example],…”
>
> In Section 5.4, I could not understand how the hyperparameters are determined in the proposed method.
> We can be more clear here.  We propose to change the wording in section 4.5.4:
>
> “For our algorithm, we could not conduct a large hyperparameter study on (beta,gamma,zeta) for computational reasons, so we instead performed cross-entropy loss training on the first training subset for 10 epochs and adjusted the values of (beta,gamma,zeta) until the other 3 loss components Lcov, Lcs, and Lproto had similar magnitude to Lce at epoch 10.  Our goal was to ensure that after the 10 epoch warmup period the relative scales of the different losses were similar and that the full training could optimize the Chebyshev Prototype Risk (CPR) while still fitting the data.  Our results indicate that our hyperparameter choices indeed allowed the model to fit the data to 100\% training accuracy while still reducing the CPR.”
>
> In Section 4.3, Eq.11 -> Eq.9?
> Yes, thank you this is a typo.  We will update this.
>
> We have also overhauled the references to make sure they are complete.
>
> If I understand correctly, the DS part in Eq.(6) can be any positive variable. Then what is the reason for using DS here?
> Moreover, the authors claim to regularize DNN training by increasing DS (which is established by decreasing DS), because it leads to a smaller value of the right part of Eq.(6). However, the larger DS value leads to the looser condition from the point of view of the left part of Eq.(6).
>
> We can put any value for 'a' into the proof of the probability bound in eqn 22, but because the left hand side | CS(v,p) - E[CS(v,p)] | only exists on the interval [0,1], any 'a' value above 1 is not useful since the probability distribution goes to zero.

---

> > ### Comment · Reviewer_wkG8 · 2023-11-20
> > **Thanks for your responses**
> >
> > Thanks for your responses.
> >
> > I agree that it makes no sense to use $a > 1$ in Eq.(22). But it is still not clear why DS is used as $a$.
> >
> > I would like to keep my score.

---

> > > ### Author Response · Authors · 2023-11-21
> > >
> > > Hello reviewer wkG8,
> > > We apologize, we did not fully understand your question initially – we should go beyond the numerical side to answer your question.
> > >
> > > Assume there are K classes and K prototypes, one for each class.  Assume that if a feature vector, v, has high cosine similarity to a prototype feature vector, p_k, from a class k, it will be classified into class k.  Likewise, for a given class and a sufficiently trained network, E[CS(v,p)] should be on the higher end of the interval [0,1] since we expect the general population of class k feature vectors to be classified correctly and similar to p_k.
> > >
> > > If a new example v were to deviate too far from its true prototype, it would get close to some other untrue prototype and be classified into that untrue prototype’s class (a misclassification).  Therefore, the further that v deviates from its true prototype, the risk of misclassification increases because it would be getting closer to other untrue prototypes.  By saying something mathematical about the probability of v deviating a certain distance from its true prototype, we are also saying something about its probability of misclassification.
> > >
> > > Now, we find its best to think in terms of “distances” (really cosine distance) between vectors and prototypes when looking at the left-hand side.
> > >
> > > The left-hand side of the inequality is | CS(v,p) – E[CS(v,p)] |.    As this difference gets larger, this means v is getting further away from its true prototype (a larger cosine distance) as compared to the rest of the feature vectors in its class.  At the same time, it is getting closer to one or more of the other (K-1) prototypes.  DS represents the average cosine distance between prototypes.  We can think about DS like a threshold distance.  If  | CS(v,p) – E[CS(v,p)] |  becomes greater than DS, then it means that v has moved a cosine distance DS away from its true prototype, but this is the average distance between prototypes from different classes!  This means that v would now very likely be close to an untrue prototype and be misclassified.  Therefore, by bounding | CS(v,p) – E[CS(v,p)] | > DS, we are bounding the probability of v transitioning from its true class prototype to an untrue class prototype and being misclassified.
> > >
> > > Pr [ distance away from true class > misclassification threshold distance  ]   <   intra-class feature covariance /  (misclassification threshold distance^2)
> > >
> > > We see that, for our learning method and intuitively, it is beneficial to increase the spacing between prototypes as much possible (increase DS) so that a feature vector must move further away from its true class to reach the misclassification threshold.  Increasing DS decreases the upper bound, which is our desired behavior.

---

> > > > ### Comment · Reviewer_wkG8 · 2023-11-22
> > > > **Thank you for your responses**
> > > >
> > > > Thank you for your responses.
> > > >
> > > > If I understand correctly, your loss $\mathcal{L}_{CS}$ is introduced to increase DS, and the justification for this comes from Lemma 3.1.
> > > > However, the inequality in Lemma 3.1 holds for any value of DS. Therefore, there is no reason why it is better to increase DS.

---

> > > > > ### Author Response · Authors · 2023-11-22
> > > > >
> > > > > Hi wkG8,
> > > > >
> > > > > It seems we may have run out of discussion period - but it has been nice to interact with you and thank you for putting in so much effort.
> > > > >
> > > > > Yes! you are right Lcs is introduced to increase DS, the spacing between class prototypes.
> > > > >
> > > > > The key is that we don't just want the inequality to hold, we want it to compress the probability down as much as possible by lowering cov/DS^2.
> > > > >
> > > > > To summarize our previous discussion, the left-hand side is the probability that I draw a new example feature vector v and its relative similarity to its prototype is greater than DS.  This could be True OR False with some probability.  The right hand side is telling us the upper limit on the probability of it being True.  Therefore, if we lower this upper limit, we compress the probability of the left hand side being True lower, which is what we want.  We do not want | CS(v,p) - E(CS(v,p))| > DS to be true, we prefer it to be False: | CS(v,p) - E(CS(v,p))| < DS.
> > > > >
> > > > > Now imagine that we train 100 different ML models maybe with different loss functions (as we have done in our results).  All of these models will have different values of DS and cov() and thus different upper limits on that probability.  Our results show that if you explicitly optimize the right hand side, you can indeed lower the probability of that left hand side being True and thus have fewer misclassifications.

---

> > > > > > ### Comment · Reviewer_wkG8 · 2023-11-23
> > > > > >
> > > > > > I agree that decreasing the cov (the numerator of the right hand side) decreases the probability of the left hand side.
> > > > > > However, regarding the DS, I still disagree with your comment.
> > > > > > It is true that the right hand side decreases as the DS increases, but at the same time the condition in the left hand side becomes looser.
> > > > > >
> > > > > > Thank you for your kind replies.

---

### Official Review · Reviewer_nGbq · 2023-10-30

**Soundness:** 3 good
**Presentation:** 4 excellent
**Contribution:** 2 fair
**Rating:** 6
**Confidence:** 4

**Summary:**

This work subscribes itself within methods for improving generalization such as Cogswell et al.'15, Rodriguez et al. '16 and Haresh et al. '18, that seek to limit the hypothesis space by reducing the variance in either covariates or among members of a class. They agree to use an idea borrowed from the group that produced a distance-based classification and nearest-class means to use as anchors, and much in the same manner as anchors in a siamese setup. Thereby global loss components that enforce distance among these class prototypes, and locals ones that enforce class-cluster compactness, are derived. As an extension almost, the authors derive bounds on the variances around class prototypes.

**Strengths:**

The paper makes clear and persuasive arguments and the motivation leads naturally to the presented solution. It provides theoretic grounds for tailoring the loss for exploiting the two classicalideas of intra- and inter- class-cluster (for the lack of better terminology). The presented theorems and proofs check out for correctness. Benchmarks are sufficiently provided.

**Weaknesses:**

I would have liked to see a theoretical understanding of why prefer your method over the competition, beyond the simple benchmark over two usual datasets.

**Questions:**

None.

---

> ### Author Response · Authors · 2023-11-17
>
> Hello Reviewer nGbq,
>
> Thank you for your helpful comments.  We have tried to address our weakness and included a new paragraph stating the reasons we prefer our method over previous methods.  Please find the new discussion paragraph in the global response or in the revised version of paper.
>
> Thank you again.

---

### Official Review · Reviewer_tQCr · 2023-11-02

**Soundness:** 2 fair
**Presentation:** 3 good
**Contribution:** 2 fair
**Rating:** 5
**Confidence:** 3

**Summary:**

This work presents an approach to reduce overfitting and improve the test performance of DNNs. It considers the existence of an optimal prototype (featurizer) and uses Chebyshev's inequality to bound the misclassification probability, which depends on (low) intra-class variance and (high) inter-class distances in the prototype. Based on this, the authors present a new loss function and showcase its effectiveness in reducing overfitting on some image classification benchmarks.

**Strengths:**

1. The idea to use Chebyshev prototype risk is novel, interesting and theoretically grounded. The authors also present a way to make their approach scalable with number of classes and it seems effective across several settings.

1. Overall, the paper is well-written and easy to follow.

**Weaknesses:**

1. ****Discussion on a set of related works seems missing.****
- The concept of minimizing intra-class variance while encouraging larger inter-class distances seems very similar to the well-observed phenomenon of neural collapse [1]. In [1], it was observed that after training for a sufficiently long time, the final layer feature embeddings collapse to class means and form a simplex ETF structure. The classifier of top also coincides with these. It was also shown to improve test performance. How does the proposed approach relate to this? I suggest including some discussion on the connection/comparisons with this.
- It seems that the section on related work on methods aimed to reduce overfitting only contains relatively older papers. For instance, [2] is a recent work that is not discussed.

2. ****Limited evaluation.****
- The proposed approach seems promising but it would be helpful to see more evidence that it is effective, e.g. by evaluating this approach on other datasets such as ImageNet.
- I would also suggest comparing with some other methods. For instance, the recently proposed squentropy loss [3] is shown to improve test performance.

****References:****

[1] V. Papyan et al., Prevalence of neural collapse during the terminal phase of training, PNAS, 2020. https://www.pnas.org/doi/10.1073/pnas.2015509117

[2] B. O. Ayinde et al., Regularizing Deep Neural Networks by Enhancing Diversity in Feature Extraction, TNNLS, 2019. https://ieeexplore.ieee.org/abstract/document/8603826?casa_token=94lYsTy6k-kAAAAA:ciG1-MsnzN_6BQRrLMz3V5PGAVLi4JB_j-EwRfsFRT-D_K9H82Cm08VspCUnM-SFvid176-wzw

[3] L. Hui et al., Cut your losses with squentropy, ICML 2023. https://proceedings.mlr.press/v202/hui23a.html

**Questions:**

(See weaknesses above)

Can the authors verify whether the baseline numbers used for comparison are computed in the terminal phase of training where neural collapse happens (please refer to [1])? Since the proposed approach seems to have a similar motivation as this phenomenon, it would be interesting to see how much incorporating the loss helps compared to just training for a large enough time.

---

> ### Author Response · Authors · 2023-11-17
>
> Hello Reviewer tQCr,
>
> Thank you for taking the time to read our paper in-depth and provide constructive comments.  We have taken your feedback and have developed some additional paragraphs to add to the paper the address your comments.
>
> Thank you for pointing us towards the line of research on neural collapse phenomena, after reviewing papers in this area, it is certainly worth a paragraph discussion in our paper because we believe our basic idea behind the Chebyshev bound development contributes a different, probabilistic mathematical model to this discussion.
>
> The main question regards whether our baseline models are in the terminal phase of training, implying that 100% of the training data has been fit and Tr(SwSb), a measure of intra-class feature covariance as a noise-to-signal ratio, has collapsed during the natural course of training (the phenomenon).  The short answer is that our baseline models have not reached the terminal phase yet and our algorithm is able to significantly reduce intra-class feature covariance compared to the baselines.
>
> We do have data on the intra-class feature covariance (numerator of CPR) for our models at epoch 100 showing that our algorithm reduced the class average numerator by 50% compared to baselines in both weight decay settings and both datasets.  Being able to reduce these numbers implies that the intra-class feature covariance had not collapsed yet to the point we could no longer effectively reduce them.  There are two limitations to comparing directly to Papyan et. al [1].  First, we use standard flipping/cropping while they did not include data augmentation, which introduces more variation in the input space, greatly benefits test accuracy, and is more standard practice in visual tasks.  Second, our intra-class feature covariance is prototype weighted because our mathematical model required it – note how our numerator is E[pi(vi − pi)pj (vj − pj )], not the typical covariance E[(vi − pi)(vj − pj )].  Having said this, we have added the following paragraph to the paper after the CPR definition:
>
> •	Our probabilistic model for CPR has parallels to Neural Collapse property (NC1), which is the collapse of within-class feature covariance when a neural network has been trained for a sufficient number of epochs beyond 100\% training accuracy (see \citep{Papyan_2020} for details).  Through this lens, our work shares two connections to Neural Collapse.  First, our results show that our algorithm significantly reduces intra-class feature covariance compared to baseline models within a typical duration (100 epochs) of training, which is not in the terminal phase of training – in this way our loss components accelerate the covariance effects of neural collapse, which has been shown to improve generalization in many settings and lends credence to our Chebyshev model that reducing CPR reduces overfitting \citep{Papyan_2020}.  The second connection is that we introduce a new probabilistic modeling of the similarity between an example’s features to its class mean feature vector (Lemma \ref{lem:2}), which to our knowledge has not been mathematically derived in the existing Neural Collapse discussion.  Future work could look to integrate theoretical arguments like ours into modeling Neural Collapse phenomena.
>
> [1] V. Papyan et al., Prevalence of neural collapse during the terminal phase of training, PNAS, 2020. https://www.pnas.org/doi/10.1073/pnas.2015509117

---

> > ### Comment · Reviewer_tQCr · 2023-11-23
> >
> > I thank the authors for the detailed response and appreciate the discussion on related works added in the paper. Overall, if the authors can include some additional experimental results for comparisons with other methods in the final version, then I am not opposed to accepting the paper.

---

### Author Response · Authors · 2023-11-17

We thank all the reviewers for their in-depth comments that will certainly make the paper better.

We have incorporated all of your comments into the paper and uploaded a revised version.  We crafted word changes and discussion points for the paper based on your comments (which we think are great additions – thank you).   We will provide responses that answer your questions and highlight where additions were made to the paper.

In addition to designing an efficient algorithm that mitigates overfitting explicitly based on our CPR bound, we want to emphasize that we view our theoretical framework as an important contribution to the discussion on regularization and overfitting.  To emphasize this, we changed the wording in our first contribution.

In this paper, we make the following contributions:

•	From: “We derive Chebyshev and Chebyshev-Cantelli probability bounds on the deviation of cosine similarity between the features of an example and its class prototype.”

•	To: “A theoretical framework based on Chebyshev probability bounds under which regularization and related training techniques can be analyzed.  The bound admits a new optimizable metric called Chebyshev Prototype Risk (CPR), which bounds the deviation in similarity between the features of an example and its true prototype.”

There are quite a few previous works on augmented loss functions and regularizers that address feature variance and/or feature separation such as [2].  We propose adding a paragraph to address these previous works and providing the reason we selected OrthoReg and DeCov specifically.  We also add a segment on the differences and advantages of our approach over previous approaches.

Page 8-9:

“There is a rich literature on regularization algorithms that optimize feature decorrelation or separation in one or more layers of a model such as (Hui et al., 2023; Deng et al., 2022; Ayinde et al., 2019; Huang et al., 2018; Pereyra et al., 2017; Liu et al.,
2016; Wan et al., 2013; Hinton et al., 2012).  Almost all of these methods work off the notion that feature decorrelation and separability are beneficial to the test performance of a network.  Our objective is to unify these approaches under a common mathematical model that details why these behaviors are desired.  By doing this, our CPR metric provides the relative numerical importance between intra-class feature covariance (weighted by prototype activations) and inter-class separation (quadratic in prototype dissimilarity).  While we wanted to show that explicitly reducing CPR in our algorithm improves overfitting over an unregularized baseline, we critically wanted to show that other non-CPR regularization techniques still implicitly improved CPR and resulted in better test performance, which would confirm our probabilistic model’s relationship to misclassification (reducing the CPR for each class reduces misclassification risk).  We chose OrthoReg and DeCov because they are simple, effective, and address feature decorrelation differently by either the weights (OrthoReg) or the activations (DeCov).  It would be worthwhile to evaluate all the methods in a similar training regime and compare their CPR metric.
We highlight some important differences and advantages of our method. First, our probability model informs us on the specific mathematical forms of intra-class covariance vs. class separation. Our feature covariance is uniquely prototype-weighted since we use $\E[p_i(v_i-p_i)p_j (v_j-p_j)]$.  All categories have differently shaped prototype feature vectors and thus our weighted covariance will scale each feature gradient differently in backpropagation depending on its relative importance to each class.  We further wanted to design a covariance algorithm that reduces the computational complexity.  Using our feature sorting and padding algorithm, our method computes the covariance contributions for a sample in $\mathcal{O}(J + JlogJ)$ time, in comparison to previous approaches in $\mathcal{O}(J^2)$ time. Our results show that over the course of training, this method still reduces model feature covariance even though we do not explicitly compute the full covariance matrix.  This gives our algorithm a distinct scaling advantage.”

Please see revised paper for full list of new references.

[2] B. O. Ayinde et al., Regularizing Deep Neural Networks by Enhancing Diversity in Feature Extraction, TNNLS, 2019. https://ieeexplore.ieee.org/abstract/document/8603826?casa_token=94lYsTy6k-kAAAAA:ciG1-MsnzN_6BQRrLMz3V5PGAVLi4JB_j-EwRfsFRT-D_K9H82Cm08VspCUnM-SFvid176-wzw

---

### Meta-Review · Area_Chair_CzwG · 2023-12-10

**Metareview:**

Although the authors are to be commended in their efforts to improve the paper during the discussion period, the paper didn't receive enough support for acceptance from the reviewers, and it still needs some improvement, as noted in the reviews and discussion. One fundamental issue is the close relation with a significant, recent amount of work about "neural collapse". Another paper that predates this and is relevant to the submission is "The role of dimensionality reduction in classification", AAAI 2014. This considered the problem of jointly training a feature extractor and classifier by making explicit the intermediate extracted features. It gave an intuitive argument for a phenomenon where, upon optimal joint training, points from the same class collapse onto "class prototypes", while the prototypes separate maximally onto a certain geometry (eg a simplex if there are more features than classes), and demonstrated it empirically. This has later been called "neural collapse" in the context of neural networks, which can often be naturally seen as consisting of a (deep) feature extraction part followed by a classifier. The authors should discuss their work in relation to these previous ideas.

**Justification For Why Not Higher Score:**

See metareview

**Justification For Why Not Lower Score:**

N/A

---

### Decision · Program_Chairs · 2024-01-16

Reject